# Improving Performance of Salt-Grown Crops by Exogenous Application of Plant Growth Regulators

**DOI:** 10.3390/biom11060788

**Published:** 2021-05-24

**Authors:** Md. Quamruzzaman, S. M. Nuruzzaman Manik, Sergey Shabala, Meixue Zhou

**Affiliations:** 1Tasmanian Institute of Agriculture, University of Tasmania, Prospect 7250, Australia; md.quamruzzaman@utas.edu.au (M.Q.); smnuruzzaman.manik@utas.edu.au (S.M.N.M.); sergey.shabala@utas.edu.au (S.S.); 2International Research Centre for Environmental Membrane Biology, Foshan University, Foshan 528000, China; 3College of Agronomy, Shanxi Agricultural University, Taigu 030801, China

**Keywords:** plant hormone, salinity stress, PGRs, wheat, gene mechanism

## Abstract

Soil salinity is one of the major abiotic stresses restricting plant growth and development. Application of plant growth regulators (PGRs) is a possible practical means for minimizing salinity-induced yield losses, and can be used in addition to or as an alternative to crop breeding for enhancing salinity tolerance. The PGRs auxin, cytokinin, nitric oxide, brassinosteroid, gibberellin, salicylic acid, abscisic acid, jasmonate, and ethylene have been advocated for practical use to improve crop performance and yield under saline conditions. This review summarizes the current knowledge of the effectiveness of various PGRs in ameliorating the detrimental effects of salinity on plant growth and development, and elucidates the physiological and genetic mechanisms underlying this process by linking PGRs with their downstream targets and signal transduction pathways. It is shown that, while each of these PGRs possesses an ability to alter plant ionic and redox homeostasis, the complexity of interactions between various PGRs and their involvement in numerous signaling pathways makes it difficult to establish an unequivocal causal link between PGRs and their downstream effectors mediating plants’ adaptation to salinity. The beneficial effects of PGRs are also strongly dependent on genotype, the timing of application, and the concentration used. The action spectrum of PGRs is also strongly dependent on salinity levels. Taken together, this results in a rather narrow “window” in which the beneficial effects of PGR are observed, hence limiting their practical application (especially under field conditions). It is concluded that, in the light of the above complexity, and also in the context of the cost–benefit analysis, crop breeding for salinity tolerance remains a more reliable avenue for minimizing the impact of salinity on plant growth and yield. Further progress in the field requires more studies on the underlying cell-based mechanisms of interaction between PGRs and membrane transporters mediating plant ion homeostasis.

## 1. Introduction

Salinity is one of the major abiotic stresses affecting crop plants and limiting production worldwide [1,2]. Globally, approximately 1125 million ha of land is affected by soil salinity [3]. Soil salinization is increasing at a rate of ~3 ha/min [4], and now becoming a major concern for the irrigated agriculture [1,5]. Salinity stress induces a multitude of responses in plants at various levels of plant structural organization. The three primary constraints imposed by salinity on plants are osmotic stress, ionic disbalance/toxicity, and oxidative stress [1]. Osmotic stress decreases external water potential and leads to a reduced water uptake capacity of plants, thus affecting cell expansion growth. It also leads to stomata closure, reducing the plant’s ability to assimilate CO_2_. The ionic stress is caused by an excess uptake of toxic salt ions (mainly Na^+^ and Cl^−^) that hamper normal metabolic processes in plants. The accumulation of toxic Na^+^ and Cl^−^ is also accompanied by a massive reduction in cytosolic K^+^, with numerous implications for a cell’s metabolic activity and viability [6,7,8,9]. Cl^–^ toxicity is less drastic compared with Na^+^, but nonetheless can cause a significant disturbance to many physiological and biochemical processes in sensitive species [10,11]. Salinity stress also leads to the production of excess amounts of reactive oxygen species (ROS) in plant tissues [12,13,14], including superoxide anion (O^2−^), hydrogen peroxide (H_2_O_2_), singlet oxygen (^1^O^2^), and hydroxyl radical (OH^●^). These ROS can severely damage the plant’s cellular structures and macro molecules like DNA, enzymes, and lipids [15,16,17,18]. Also, ROS are highly potent regulators of a broad range of Ca^2+^, Na^+^, and K^+^—permeable ion channels [19,20,21]—thus causing a major disturbance to stress signaling and intracellular ion homeostasis, well before damaging effects become evident.

Three major cereal crops, namely wheat, rice, and maize, are responsible for over 50% of daily caloric uptake by the human population. All of them are classified as salt-sensitive species and perform poorly when grown on saline soils. For example, wheat provides about 20% of human food energy requirements and 25% of proteins consumed daily worldwide (Wheat Initiative, www.wheatinitiative.org/, accessed on 23 May 2021). Wheat is a salt-sensitive glycophyte [22], and salinity is considered to be a major soil constraint in the Australian Wheatbelt [23,24] which results in about a 40% yield reduction [25], costing Australian economy ~A$200 million per annum [26].

Two different (but potentially complementary) approaches can be used to reduce the negative impact of salinity stress on plant growth and yield. The first one is the development of salt-resistant cultivars via molecular or classical breeding. The second approach is related to agronomical means, and includes inoculating seeds with halotolerant plant growth promoting rhizobacteria (PGPR) or the application of various plant growth regulators (PGRs) [27,28,29,30,31,32,33,34]. While genetic improvement is considered as the best solution from a long-term perspective, no significant progress has been made in breeding programs. This is due to the polygenic nature of tolerance, which reflects the complexity of salt tolerance mechanisms in plant, and the lack of available genes that confer salt stress tolerance [35,36,37]. Conventional breeding techniques are time-consuming and laborious, and have met with only a limited success. With the advancement of science and technology, molecular techniques and transgenic technology have been widely used in plant breeding worldwide. Although transgenic technology is considered as a fast and effective method to obtain salt-tolerant varieties, the public acceptance of genetically modified (GM) crops remains a major stumbling block in most countries [38,39,40]. In this context, PGPRs could potentially minimize the detrimental effects of salinity stress on plant growth and yield without triggering these public/governmental concerns. PGRs share the common function of regulating intrinsic hormone levels within plants by modulating signaling within various hormone transduction pathways, and are widely available and easy to apply to crops [41,42]. However, the effectiveness of PGPRs depends upon their interaction with host plant and soil environment. Sometimes plant growth-promoting bacteria has exhibited harmful effects on the growth and development of plants [43], and is often considered to be “unsafe” for human and animal health [44]. Also, PGRs cross-talk with each other and may act synergistically or antagonistically to regulate plant growth, development, and defense responses, generally by inducing gene expression [45]. This complexity may result in a certain level of unpredictability and negate expected beneficial effects. The main aim of this review was to summarize the bulk of the reported data on the use of PGRs for improving performance of plants grown on salt-affected lands, revealing underlying cellular mechanisms and downstream targets, and critically assessing the applicability of PGR for sustainable crop production under conditions of soil salinity.

## 2. Plant Growth Regulators

Plant growth regulators (PGRs) are defined as synthetic or naturally occurring organic compounds that influence biological process in higher plants at very low concentrations [41]. PGRs can increase or decrease growth and development by altering their normal biological processes [41,46,47]. When these compounds occur naturally inside the plant they are known as phytohormones, but when applied exogenously they are called PGRs [48]. PGRs act as signaling agents, allowing plants to maintain plasticity during growth and development, and are hence considered as principal factors in responses of plants to biotic and abiotic stresses [49,50]. PGRs play a significant role in alleviating salt stress via a broad range of physiological and developmental alterations [51,52,53]. In broad terms, PGRs are involved in increasing the physiological availability of water and essential nutrients, while helping plants reduce toxic salt load [54]. They also have a major impact on the antioxidant enzyme activities in plants [55,56,57,58,59,60,61,62]. PGRs induce salt tolerance by increasing the activity of ROS scavenging enzymes to maintain the ROS at a nontoxic level under stress conditions [63,64]. The ameliorating ability of PGRs depends on environmental factors that affect their absorption, the concentration at which they are applied, and the physiological state of the plant [65,66,67]. PGRs are classified into a number of distinct classes, such as auxins (Aux), cytokinins (CKs), nitric oxide (NO), brassinosteroids (BRs), gibberellins (GAs), salicylic acid (SA), abscisic acid (ABA), jasmonates (JAs), and ethylene. The following sections examine the efficiency and modes of ameliorating the effects those PGRs have on how plants respond to salinity stress.

## 3. Effects of Plant Growth Regulators on Plant Performance under Saline Conditions

### 3.1. Auxins

Auxin is a widely used plant growth regulator of low molecular weight with an aromatic ring structure [68]. Auxin is produced in growing shoot tips and transported down the main stem via the polar auxin transport (PAT) mechanism [69]. It is involved in cell division and elongation, organogenesis, and apical dominance [70,71,72,73,74]. Auxin stimulates cell elongation via increasing wall extensibility and participates in the regulation of cell wall properties by inducing wall loosening [75]. Auxins also stimulate H^+^–ATPase operation, creating a driving force for inorganic ion uptake that contributes to increased cell turgor [76,77].

Salt stress affects the growth of both primary and lateral roots [78,79,80]. The developmental plasticity of the plant root under saline conditions is regulated by auxin [81], and the inhibition of root growth is associated with reduced auxin accumulation [82], most likely via the PAT mechanism [83]. Slowing root growth might be an adaptive mechanism for plants surviving in salt environments. Exogenous application of auxin leads to an increase in root growth under saline conditions in several species [82,84]. However, this increase in root growth comes with a carbon cost, and therefore, could be counterproductive under conditions of severe salinities, when (limited) plant ATP pull is required for a range of defense responses. Lateral root development is also an important survival strategy for plant to avoid damage in unfavorable environmental conditions, including salt stress [85,86,87]. Plants remodel their root architecture by altering auxin accumulation and its redistribution under salt stress conditions [88,89,90,91].

Indole-3-acetic acid (IAA) is an important member of auxin family of plant hormones [92,93]. Stress conditions lead to a significant reduction in IAA concentrations in rice, maize, tomato, and wheat plants [94,95,96,97]. This reduction of free IAA concentration in crop is cultivar- or plant organ-dependent. For example, the free IAA concentration remained constant in the roots of a salt-tolerant maize hybrid line, but significantly decreased in the roots of a salt-sensitive line, while IAA concentration in leaves remained constant in both lines [95]. Therefore, the reduction in plant growth and development under stress conditions could be an outcome of altered auxin accumulation and redistribution. Consistent with this, some studies suggested that exogenous application of Aux may alleviate salinity stress in many crops, including wheat [45,84,98,99]. For example, foliar spraying of 2 mM IAA to salt-grown maize plants increased kernel yield by ~9%, while a combination of IAA spray and a basal application of inorganic nutrients (K and P) improved yield by 20% under 10 dS m^−1^ saline conditions [98]. In rice, a foliar spray of IAA (50 mL per pot) at the reproductive stage increased grain yield by ~4% and ~46% in salt-tolerant and salt-sensitive cultivars, respectively, under 6 dS m^−1^ saline conditions [99]. In the case of wheat, seed primed with 100, 150, and 200 mg L^−1^ of different synthetic auxins, namely 4-dichlorophenoxyacetic acid (2,4-D); 2,4,5-trichlorophe-noxyacetic acid (2,4,5-T); and α-naphthaleneacetic acid (NAA) improved the performance of salt-grown plants under both controlled environment and field conditions. However, the effect was cultivar-, priming agent-, and concentration-dependent, questioning the direct causal link between PGR application and observed responses. Seeds primed with 150 mg L^−1^ NAA increased plant grain yield by ~36% and ~27% in salt-sensitive and salt-tolerant wheat varieties, respectively, when grown under 150 mM saline conditions [100]. The use of IAA-producing bacteria also lead to an improvement of wheat performance under saline conditions [101].

One of the modes of action for the auxin-driven decrease in the extent of salt damage on plant performance is increasing the activities of auxin-responsive genes that increase the plant’s water retention capacity and decrease H_2_O_2_ accumulation [45,102]. The auxin responsive *TaIAA* gene family exhibits differential expression during the absorption of inorganic salts in wheat roots [103], and an auxin responsive gene small auxin-upregulated RNAs *TaSAUR75* in wheat regulates plant growth and development under saline conditions by preventing H_2_O_2_ accumulation under salt stress [102]. The salt-responsive gene *TaEXPB23* is downregulated by the exogenous application of IAA, and overexpression of *TaEXPB23* conferred salt stress tolerance by decreasing osmotic potential and enhancing the water retention ability in transgenic tobacco [45].

### 3.2. Cytokinins

Cytokinins (CKs) are an important class of phytohormones that promote cell division in the roots and shoots of plants. There are two types of CKs: the adenine type and the phenylurea type [104]. To date, phenylurea CKs have not been found in plants [105]. The adenine-type plant hormones are the main form and classified as isopentenyladenine (iP), trans-zeatin (tZ) and *cis*-zeatin (cZ), and dihydrozeatin and its riboside [106,107]. The iP-type cytokinin in the phloem is transferred from aerial parts of the plant to the root to maintain vascular structure in the root meristem [108]. The tZ-type cytokinin is transferred from the root to aerial parts of the plant through the xylem to regulate shoot growth [109]. As a result of these redistributions, CKs participate in various biochemical and physiological processes, such as cell division and leaf senescence, thus controlling the root/shoot ratio. The long-distant CK transport is also essential for plant responses to abiotic stresses, including salt [110,111] (Figure 1).

CK has both a positive and negative regulatory role in alleviating the detrimental effects of salinity stress. For example, cytokinin deficiency leads to enhanced salinity tolerance in *Physcomitrella patens* and *Arabidopsis* [111,112,113], and contributed to yield improvement in many crops [114]. Cytokinin oxidase (CKX) is the prime enzyme involved in CK metabolism, which can effectively reduce the concentration of CK in plants. The overexpression of CKX significantly influences hyposensitivity to salt stress in *Physcomitrella patens* [111,112]. Regulation of isopentenyltransferase (IPT) genes under conditions of salinity stress also reduces CK content in *Arabidopsis* and enhances its tolerance to salinity [112,115,116]. On the other hand, overproduction of CKs by inducible expression of *IPT8* gene leads to reduced adaptive ability in *Arabidopsis*. The excess production of CKs decreases transcript levels of ROS scavenging enzymes, thus leading to increased ROS production that correlates with sensitivity to salt stress [113]. In contrast to the above negative effects of CKs, other studies have indicated a beneficial role of CK for plant performance under salinity stress. The application of CKX inhibitor INCYDE protected the photosynthetic apparatus and increased the production of flowers in tomato plants [117]. The downregulation of CKX2 under saline stress condition significantly increased CK concentration and reduced yield penalty in rice [118,119]. AGO2 (argonaute RISC catalytic component 2) plays a key role in improving salinity tolerance by changing the level of CKs and enhancing grain yield in rice [120]. These contradictory reports question the practical application of CKs, and suggest concentration- and tissue-specific modes of action.

Published papers indicate that salt stress conditions are often associated with a decrease in CK concentration in crops like rice and wheat. CK spraying leads to an increase in grain yield under saline conditions [121]. Exogenous CKs may increase plant antioxidant enzymes activity and reduce ROS load. Kinetin (adenine-type CKs) spraying at a rate of 10 mg L^−1^ ameliorated the deleterious effects of salinity by reducing the uptake of toxic ions Na^+^ and Cl^−^ and promoting the uptake of K^+^ in wheat seedlings. Seed primed with different concentrations of synthetic cytokinins (kinetin and benzylaminopurine) increased grain yield by up to 52%, but in a strong cultivar- and concentration-dependent manner [60].

CKs mainly promote physiological responses through the regulation of gene expression [122,123], but little information is available about the molecular function of cytokinin under salt stress conditions. In *Arabidopsis*, CRF6 (cytokinin response factor 6) represses cytokinin-associated genes during oxidative stress [124]. A salt-inducible novel wheat *TaCKX3* (cytokinin oxidase/dehydrogenase) gene is located on chromosome 7B [125], and silencing of the *TaCKX1* gene increased grain yield in wheat [126]. The expression of the high-affinity potassium transporter *AtHKT1.1*, which controls xylem Na^+^ loading, was repressed by CK treatment in *Arabidopsis* [127]. The genes involved in ROS breakdown are also greatly affected in cytokinin-deficient mutant *ipt1,3,5,7* [113]. CKs help induce cytokinin response factors (CRFs) in the ERF-VI subfamily; CRFs positively regulate osmotic stress tolerance [128,129]. However, other reports have suggested that overproduction of CK resulted in a negative effect in plants by modulating stress-responsive gene expression. For example, overexpression of cytokinin biosynthetic gene *AtIPT8* (adenosine phosphate–isopentenyl transferase 8) significantly inhibits true leaf emergence and primary root growth under salt stress conditions. It is also associated with increasing ROS production, decreasing survival rates, and chlorophyll content, which lead to reduced salinity tolerance [113]. These pleiotropic effects question the practicalities of CK application and suggest that balancing CK levels is essential for adapting plants to salt conditions.

### 3.3. Nitric Oxide

Nitric oxide (NO) is a gaseous free radical that acts as a signaling molecule. NO synthesis in plants is mainly carried out by L-arginine-dependent, nitric oxide synthase-like activity and nitrate reductase (NR)-catalyzed reduction of nitrite (NO_2_^−^) [135,136,137,138]. It is widely known as a “jack-of-all-trades” in stress responses [139,140]. Exogenous application of NO enables plant protection against various abiotic stresses, including salinity [141,142,143,144]. For example, exogenous NO treatment increases K^+^ concentration and decreases Na^+^ concentration in salt-grown plants, thus maintaining an optimal K/Na ratio that is critical for plant’s operation [145]. NO alleviates osmotic stress by scavenging reactive oxygen species through increased antioxidant enzyme activity [141,146,147,148] and glucose-mediated repression of photosynthesis [148]. NO protects the mitochondria from oxidative damage by increasing ATP synthesis, and seed priming with NO increases wheat grain yield by up to 22% under saline conditions [147]. NO also significantly modulates both H^+^–ATPase and H^+^–PPase (H^+^–pyrophosphatase) activities, thus conferring salinity tolerance in plants [149]. These results suggest that application of exogenous NO could potentially improve crop growth and development under salt stress conditions.

Despite the above beneficial reports, the practical application of NO in ameliorating the detrimental effects of salinity is questionable for several reasons. First, similar to the cytosolic Ca^2+^ signaling, stress-induced elevation in NO levels is usually transient and requires a return to the basal level. Second, NO is involved in multiple signaling pathways, so alteration in basal NO level caused by its exogenous application may interfere with some of them. Last but not least, the biological lifetime of an NO molecule is relatively short (millisecond range [150]), questioning its ability to sustain long-term control of transporters activity.

### 3.4. Gibberellins

Gibberellins belongs to a large group of tetracyclic diterpenoid carboxylic acid derivatives which have various physiological functions, such as stimulating organ growth through enhancement of cell division and cell elongation [151,152]. Gibberellic acid (GA) is the most common form of gibberellin [153,154,155]. The biosynthesis of GA is regulated by both developmental and environmental stimuli [156]. Salinity stress reduces endogenous GA content, resulting in plant’s hypersensitivity to salt [157,158].

DELLA family protein is a major GA-negative regulator that may be involved in different environmental and hormonal signaling. For example, DELLA protein SLR1 plays a role in inhibiting plant growth by inhibiting GA signaling under salt stress conditions [159]. Overexpression of some other GA catabolism-related genes like *OsGA2ox5* [160] and *OsMYB91* [161] in rice and *AtGA2ox7* [162] in *Arabidopsis* reduces growth and shows an enhanced tolerance to salt stress compared to wild plants. Again, OsCYP71D8L is a potential GA-deactivating protein that plays a significant role in balancing the growth process and stress responses, and leads to enhanced tolerance to salt stress in rice [163]. These results suggest that the reduction of GA signaling under salt stress conditions is directly associated with salt tolerance in plants. On the other hand, some papers have reported that exogenous application of GA had a positive effect on salt stress tolerance in many crops. In this context, increasing lipid biosynthesis is one of the essential mechanisms conferring salinity stress tolerance in plants. This process is disturbed by salinity, but the exogenous application of GA leads to up-regulation of chloroplast lipid biosynthesis, which is directly associated with increasing salt stress tolerance in rice [164]. Salt stress also reduces enzyme activities, as well as hampering the nutritional balance in plants. A foliar spray of 0.1 mM GA significantly alleviates the damaging effect of salt and increases growth and enzymatic activities in okra [165]. Another potential target of GA are expansins, which determine extensibility and mechanical properties of cell walls. The *TaEXPB23* transcript expression in wheat was upregulated by salt stress but downregulated by exogenous GA application, and constitutive overexpression of *TaEXPB23* enhanced salt stress tolerance in transgenic tobacco by enhancing water retention ability and decreasing osmotic potential [45].

### 3.5. Brassinosteroids

Brassinosteroids are primarily polyhydroxylated, sterol-derived plant growth regulators. They are ubiquitous in all plant species, and are implicated in a wide range of growth and developmental processes in various crop plants [166,167,168,169,170]. BRs are involved in the regulation of multiple physiological, developmental, and biochemical processes, including seed germination, cell division and elongation, differentiation of vascular tissues, development of root and shoots, senescence, reproduction, and photomorphogenesis. BRs are also essential for plant adaptation to various abiotic and biotic stresses [166,171,172]. BRs interact with other hormones to regulate these types of activities in plants [169]. The most common, and thus best studied, are 24-Epibrassinolide (24-EBL) and 28-homobrassinolide (28-HBL) [173,174]; studies of these have included their role in mitigating salt stress in plants [175,176].

A foliar spraying of EBL ~10^−8^ M or seed soaking with ~10^−6^ M scavenged excessive ROS through the enhancement of antioxidant enzyme activities and modified the activity of proline metabolism, thus improving salinity stress tolerance in wheat [177]. Exogenous 5 μM EBL treatment resulted in a ~42% increase in pod yield in salt-grown bean plants [178], and foliar spray of 0.125 mg L^−1^ BR led to 18–35% increase in seed yield in peas [179].

The molecular mechanisms explaining how BRs control stress responses and regulate stress-responsive gene expression in plants are largely unknown [180,181,182,183]. BRs bind to a small family of leucine-rich repeat receptor kinases (BRI1) at the cell surface, thereby initiating an intracellular signal transduction cascade that results in altered expression of hundreds of genes that are implicated for diverse functions, including increased adaptation to various stresses [184]. For example, enhancing BR signaling activity in *Arabidopsis* led to increased salt stress tolerance, but BR-defective mutants showed sensitivity to salt stress [181]. The transcript levels of the brassinosteroid receptor (*OsBRI1*) were greatly influenced by EBL and its combination with salt stress in rice. On the other hand, the salt responsive gene (*SalT*) was negligibly expressed by the combination of salt and EBL [185]. The *Arabidopsis* ubiquitin-conjugating enzyme, *UBC32*, a stress-induced functional ubiquitin conjugation enzyme, is associated with endoplasmic reticulum protein degradation (ERAD) and brassinosteroid mediated growth promotion, as well as salt stress tolerance [186]. Brassinosteroids may also enhance abiotic stress tolerance through their interaction with other plant hormones, such as ABA [187,188,189]. The crosstalk between BR and ABA occurs after BR perception, but at or before BIN2, so a large portion of BR responsive genes are also regulated by ABA [190]. The specific details of this interaction need to be investigated in future studies.

### 3.6. Salicylic Acid

Salicylic acid is a phenolic compound and important endogenous growth regulator that participates in the regulation of biotic and abiotic stress responses in plants [191,192,193]. SA has been shown to affect membrane permeability, bud growth, growth rate, stomatal closure, mitochondrial respiration, material transfer, photosynthesis, and ion absorption. Because of this, SA plays an essential role in mediating plants’ adaptive responses to salinity [52,194], and controls membrane permeability (hence, ion uptake and transport) [195]. It also maintains redox homeostasis in cells under salt stress conditions [196,197,198] (Figure 2). External application of SA was found to improve tolerance to salt stress in wheat plants as a result of the upregulation of transcript levels of *GPX1*, *GPX2*, *DHAR*, *GR*, *GST1*, *GST2*, *MDHAR*, and *GS*, and further enhancement in the enzyme activities of AsA-GSH cycle [199]. SA also reduces the extent of oxidative damage caused by salt stress by enhancing the activities of peroxidase and catalase, as well as the production of osmoprotectant compounds, such as proline, betaine and glycine [193,200]. SA regulates the transcript levels of the genes encoding ASA–GSH cycle enzymes, such as DHAR (dehydroascorbate reductase), GPX (glutathione peroxidase), GR (glutathione reductase), GST (glutathione-S-transferase), MDHAR (monodehydroascorbate reductase), and GS (glutathione synthetase) [199]. SA treatment also improves the K^+^/Na^+^ ratio in salt-grown plants [59,201].

The effectiveness of exogenous SA at mitigating salt stress damage depends on crops and the concentration of NaCl in the growing media. For example, it can reduce the negative effect of salt stress when plants are exposed to moderate stress (0.3% and 0.6% NaCl), but cannot counteract severe salt stress (0.9%) in Caryophyllaceae [202]. Both arial spraying and seed priming with 1 mM SA have improved grain yield of wheat and pearl millet by ~13–14% [203]. The effectiveness of SA also varies with the concentration of endogenous or exogenous SA. The higher accumulation of endogenous SA led to hypersensitivity to NaCl [204,205,206], while activating the SA signaling pathway significantly improves salinity tolerance. For example, overexpression of SA receptors (*MhNPR1* or *AtNPR1*) enhanced tolerance to salt/osmotic and oxidative stress by increasing SA signaling in tobacco [207,208]. On the other hand, a lack of SA receptor enhanced salt sensitivity in plants [209]. Similarly, exogenous application of three levels of SA (0.5, 1.0, and 1.5 mM) as a priming agent was evaluated in mung bean grown under different salt concentrations (3, 6, and 9 dS m^−1^). In most of the cases, a moderate concentration (1 mM SA) gave the best result in terms of ion content, gas exchange parameters, and chlorophyll content in leaves [210]. In *Arabidopsis*, the inhibitory effect of high salinity was exaggerated by >100 μM SA treatments, while plants benefited from <50 μM treatment during the seed germination [211].

### 3.7. Abscisic Acid

Abscisic acid is known as a stress hormone that mediates different types of biological and non-biological stress in plants [216,217,218]. ABA is synthesized in all de novo plant parts such as roots, flowers, leaves, and stems [219]. As an endogenous signaling molecules ABA enables plants’ survival under adverse environmental conditions, including salinity [220,221] (Figure 3).

During salt stress conditions, endogenous levels of ABA increase, which enhances plant adaptation to salinity by limiting ROS accumulation [222]. Higher accumulation of ABA also induces stomata closure, thus reducing transpiration for better water saving under osmotic stress conditions caused by salinity [223,224]. The accumulation of ABA occurs more rapidly in roots than leaves [95,225], and biosynthesis of ABA is associated with lateral root development in plants under salt stress conditions. This process is believed to be related to ABA regulation of auxin distribution under NaCl treatment. Interestingly, ABA biosynthesis inhibitor fluridone and an ABA biosynthesis mutant (*vp14*) successfully rescued the *Arabidopsis* phenotype under saline conditions [226]. Exogenous application of ABA increases the number of lateral roots in the ABA receptor mutants (*pyl8* and *pyl9*) in *Arabidopsis*, thus conferring to the plants a tolerance to salt [227]. These results indicate that the activation of ABA signaling displays enhanced salinity tolerance in crops and is triggered by an external application of ABA.

In wheat, ABA reduces salt stress damage by regulating proline content [228], and also by reducing the ROS levels in salt-grown plants [229]. Seed priming with ABA decreases Na^+^ content and increased K^+^ content in flag leaves, leading to increased number of grains per spike and grain yield of wheat under saline conditions, with up to 49% yield increase being reported [230]. In rice, exogenous application of 100 μM ABA solutions has improved plant performance by increasing *OsP5CS1* and *OsP5CR* gene expression, which triggered proline accumulation, although the effect was varietal-dependent [231]. In wheat, the spraying of a moderate concentration of ABA (50 μmol·L^−1^) improved salt tolerance, while a higher concentration (100 μmol·L^−1^) had no significant impact [232]. Sorghum leaves were fed with an exogenous ABA to control shoot Na^+^ concentration and improve plant growth. The growth enhancement and lower Na^+^ content in shoots occur at a lower ABA concentration (≈10 mmol m^−3^) than a higher ABA concentration (≈40 mmol m^−3^ or above) under 150 mol m^−3^ NaCl treatment. A higher dose of ABA is needed to adapt plants to treatment with a lethal dose of NaCl (300 mol m^−3^). It is known that ABA acts by inducing transitional stomata closure to reduce transpiration and increase water use efficiency, thus lowering transpiration; this probably plays a significant role in reducing transporting Na^+^ from root to shoot [233]. In potatoes, exogenous application of ABA improves stomatal conductance and leaf relative water content under saline conditions, but the effect varies with genotype and method of ABA application, suggesting external ABA control over stomata functioning and better water saving under saline conditions [234].

Wheat LEA (late embryogenesis abundant) protein DHN-5, induced by salt and abscisic acid, can confer salt and osmotic stress tolerance, while *Dhn-5* transgenic plants exhibit higher germination rates and leaf area, as well as better growth. The above salinity tolerance of the transgenic plant could be due to higher K^+^ accumulation in leaves and osmotic adjustment developed by active accumulation of proline [235]. In another study, a new member of the CIPK (calcineurin B-like protein-interacting protein kinase) gene family (*TaCIPK29*) has been identified in wheat. The *TaCIPK29* transcription level increased after the treatment of ABA and NaCl. *TaCIPK29* transgenic plant shows higher K^+^/Na^+^ ratio and increased activity of peroxidase (POD) and catalase (CAT) under salt stress [236]. A novel, ABA-inducible *TaSC* gene was cloned from a salt-tolerant wheat mutant, RH8706-49 [237], that operated in a CDPK pathway, enhancing intercellular K^+^/Na^+^ ratio and chloroplast function. Exogenous ABA treatment also promoted early salt stress-responding genes *WESR1* and *WESR2* in wheat [238]. A basic helix-loop-helix wheat gene (*TabHLH1*) mediates plant adaptation to osmotic stresses. This gene is associated with promoting stomata closure and increasing biomass production under salt and ABA treatment. The overexpression of *TabHLH1* enhances leaf water retention capacity in transgenic tobacco, indicating better water saving to adapt under saline conditions [239].

While exogenous modulation of ABA levels in plants comes with improved water use efficiency (WUE), the resultant stomata closure may compromise the plant’s ability to assimilate CO_2_ (hence, biomass gain). This calls into question the long-term efficacy of such approaches for field-grown crops exposed to salinity.

### 3.8. Jasmonates

Another important set of PGRs are the jasmonates. Jasmonic acid (JA) and its methyl ester (MeJ) are known as jasmonates, and control a wide range of plant growth and developmental activities, as well as adaptive plant responses to a range of biotic and abiotic stressors, including salinity [242,243]. The biological activities of JA are significantly increased when plants are exposed to excess levels of salt [244], with stronger responses from salt-tolerant cultivars. This has prompted a suggestion to use jasmonic acid content as a proxy for salinity tolerance in plants [245,246,247]. Activation of the JA signaling pathway increases the accumulation of JA and increases plant salinity tolerance [248]. Consistent with this, the JA receptor mutant is associated with greater cell elongation under saline conditions to confer salinity tolerance [249]. Exogenous application of JA alleviates the toxic effects of salt by maintaining ion homeostasis, increasing ROS scavenging enzymatic activities, and improving stomatal functioning.

In wheat, exogenous 2 mM JA treatment alleviated salt stress by enhancing the activities and transcript levels of antioxidant enzymes, such as CAT, SOD, and APX. It also boosted the content of reduced glutathione (GSH) and carotenoids, thus decreasing the peroxidation of lipids [57]. Foliar JA sprays are beneficial for improving the grain yield of salt-grown soybeans [250]. The effectiveness of JA depends on its concentration and the level of salinity in the growing media [251].

JA induces biological and non-biological stress responses through the jasmonate signaling pathway [252]. For example, a salinity-responsive bread wheat gene *TaAOC1* was constitutively expressed in both bread wheat and *Arabidopsis*, and was upregulated by exogenously supplied JA and ABA. The expression of *TaAOC1* in both *Arabidopsis* and wheat restricted root growth, but enhanced salt tolerance and JA content by increasing SOD activity, indicating that JA was involved in the orchestration of salt stress response and developmental processes [248]. Other studies have shown that *TaAOC1* and *TaOPR1* [16,253] are the two genes that provide salt tolerance via both JA- and ABA-dependent pathways to promote expression of MYC2, a crucial component of abiotic stress response-signaling pathway [254]. Large-scale transcriptomic studies have shown that some JA-biosynthesis genes (e.g., AOC1, AOC2, AOS, LOX3 and OPR3) are up-regulated in roots under salt stress [255,256,257,258]. These findings suggest that JA signaling pathway is activated by salt stress and triggers an array of physiological and growth changes in plants. The *TIFY* gene family is regulated by salt and JA treatment, and transgenic lines over-expressing *TdTIFY11a* showed higher germination and growth rates under high-salinity conditions, indicating that it acts as jasmonic acid signaling [259]. A salt-responsive wheat gene *TaEXPB23*, associated with enhanced water retention ability and decreased osmotic potential, was upregulated by JA in transgenic tobacco [45].

### 3.9. Ethylene

Ethylene is a gaseous signaling molecule known as a stress-responsive hormone in plants [52,260]. It regulates a broad array of physiological and developmental responses [261,262] by cross-talking with other signaling molecules [263,264]. The functional effectiveness of ethylene depends on the sensitivity of plants to the hormone and its concentration in the cell [265,266,267,268]. Salt stress conditions cause a rapid increase of ethylene and its direct precursor ACC (1-aminocyclopropane-1-carboxylic acid) production inside the cell [269,270]. Higher ethylene production results in salt-sensitive phenotypes in many plants, such as rice, *Arabidopsis*, pepper, lettuce, spinach, and beetroot [269,271,272]. However, other studies have shown that the overproduction of endogenous ethylene or exogenous treatment of ethylene-releasing compounds, such as ethephon or ethylene precursors like ACC, may lead to increased salinity stress [273,274,275,276]. These results suggest that the effectiveness of ethylene in mitigating plant responses to salinity is crop- and genotypic-specific, and is controlled by the concentration of ethylene in a specific cellular compartment.

Similar to other PGRs, ethylene modulates salinity tolerance by maintaining tissue Na^+^/K^+^ homeostasis and inducing the antioxidant defense system [277,278,279]. Consistent with this, exogenous application of an ethylene-releasing compound significantly improves salinity tolerance in *Arabidopsis*. Exogenous 10 µM ACC (1-aminocyclopropane-1-carboxylic acid), an ethylene precursor, suppressed K^+^ loss and enhanced Na^+^ extrusion from the root, thus maintaining K^+^/Na^+^ homeostasis during short-term NaCl treatment in *Arabidopsis* [280]. While exogenous 30 μM ethephon (another ethylene-releasing agent) confers salt stress tolerance by increasing K^+^ ion content in shoots and roots rather than decreasing Na^+^ content in *Arabidopsis*, it also recovers salt-induced reductions in root growth [273]. In wheat, seed germination, as well as root and shoot length are significantly improved by different concentrations of ethephon under 100 mM NaCl treatment [281].

Ethylene response factors (ERFs) are key regulators in abiotic stress tolerance, including salinity. The transcription of wheat *ERF* gene (*TaERF1*) is induced by salinity, and overexpression of this gene activates stress-related genes that eventually increase salt stress tolerance in transgenic plants [282]. The seedlings of the *TaERF3*-overexpressing transgenic lines exhibit significantly enhanced tolerance to both drought and salt stresses compared to untransformed wheat [283]. Overexpression of an ethylene-responsive transcription factor (TdSHN1) from durum wheat resulted in the development of a thicker cuticle and lower stomatal density, thus reducing water loss in transgenic tobacco [284]. Transcripts of the lipid transfer protein gene (*TaLTP1*) were increased by salt and ethephon treatment in wheat [285]. LTPs enhance cell membrane integrity and ROS scavenging in transgenic potatoes [286]. They also result in reduced Na^+^ accumulation in transgenic tobacco [287]. A wheat aquaporin gene *TaAQP8* conferred salt stress tolerance in transgenic tobacco by increasing the K^+^/Na^+^ ratio and Ca^2+^ content, and by reducing membrane damage and H_2_O_2_ accumulation [288]. Its transcript levels were induced by both ethylene and NaCl. Ethylene also induces many early response genes that are essential for ribosomal protein activation, chaperoning synthesis, ROS scavenging, and carbohydrate metabolites pathway [289].

## 4. Summary and Recommendations

When exposed to saline stress, plants display retarded growth and development and yield losses, and employ a range of mechanisms to deal with various constraints imposed by saline soils. Plant hormones play an important role in this process. Using exogenously applied PGRs remains a highly attractive option to plant growers, as a cost-effective method to induce salt tolerance genes and assist plants in adapting to hostile salinity conditions. However, the effectiveness of PGRs depends on the level of salt stress, genotype, timing, and methods of applications, as well as PGR concentrations. The issue is also complicated by the facts that plant hormones are involved in numerous developmental and adaptive responses (not only those related to salinity), and hormonal signaling pathways have a very significant overlap. Thus, elevation in the basal level of one of the PGRs could result in a major disturbance to some other signaling pathways, with pleiotropic effects for growth, development, and adaptation. This is specifically true for PGRs that modulate endogenous ROS and NO levels. The practical applicability of PGRs should also be considered in technological and economic contexts. Root treatment with PGRs reported in many papers is appropriate for laboratory-based studies, but has no place in the field. The aerial PGR sprays are more practical, but require significant technological developments (e.g., the use of surface surfactants, timing of spray application, etc.). The overall effects of PGR sprays will also be strongly dependent on environmental conditions (temperature, humidity, time of the day), as their penetration into the leaf will be largely determined by the extent of the stomata opening. The cost–benefit analysis of the efficacy of PGR ariel sprays should also be taken into account. We would like to illustrate the latter point by one simple example. In Yusuf et al. [177], the authors reported an 18% to 35% increase in seed yield in salt-grown peas, using a foliar spray of 0.125 mg L^−1^ of 24-epibrassinolide (EBL). The current cost of 10 mg of EBL from Sigma-Aldrich (Sigma-Aldrich Pty Ltd., NSW, Australia) is $588, and will be sufficient to make only 80 L of solution. The typical field rate of aerial spray application is 450 L ha^−1^ [290], so the cost of spraying of 1 ha will be about $3300 (EBL only). At the same time, the “target benchmark” for pea production in Australia is 8 tons ha^−1^ [291], with the commodity price being around $1000 per tonne in 2019 [292]. Thus, even a 30% increase in yield following EBL application will only result in a benefit of $2400 ha^−1^, which is clearly not enough to cover the cost of EBL application. The same logic is applicable to all other PGRs. Thus, all above beneficial reports of PGR application need to be taken with a “pinch of skepticism” and critically evaluated for their economic rationale.

In this context, we believe that future progress in the field may be achieved not by exogenous application of PGRs, but rather by understanding a causal link between PGRs and their downstream effectors mediating plants’ adaptation to salinity, and then incorporating these findings into a variety of plants via molecular breeding. This task, however, remains a great challenge, and can be only resolved by moving from whole-plant studies (employed by 95% of published papers) to more in-depth studies at the cellular level, using a modern range of biophysical and imaging techniques that allow quantification of the operation of key transport systems conferring plant ionic and oxidative homeostasis under stress conditions.

## Figures and Tables

**Figure 1 biomolecules-11-00788-f001:**
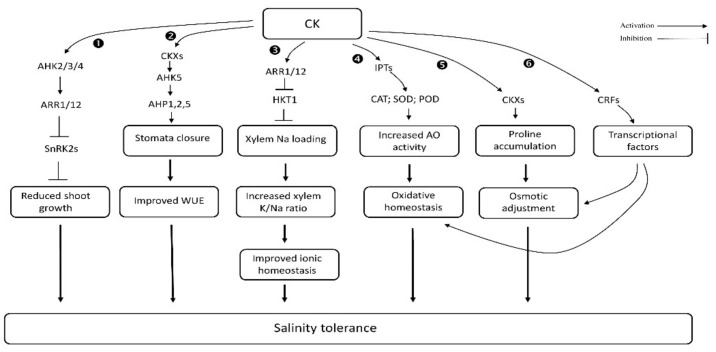
Effect of cytokinin on salt stress tolerance in plants (based on ❶ [115,130], ❷ [131], ❸ [127,132], ❹ [133], ❺ [134], and ❻ [128,129]).

**Figure 2 biomolecules-11-00788-f002:**
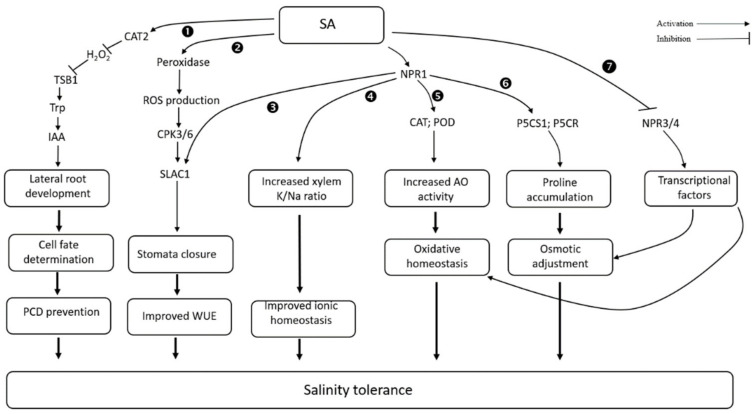
Effect of salicylic acid on salt stress tolerance in plants (based on ❶ [212], ❷ [213], ❸ [214,215], ❹ [52], ❺ [193,200], ❻ [52,207], and ❼ [207]).

**Figure 3 biomolecules-11-00788-f003:**
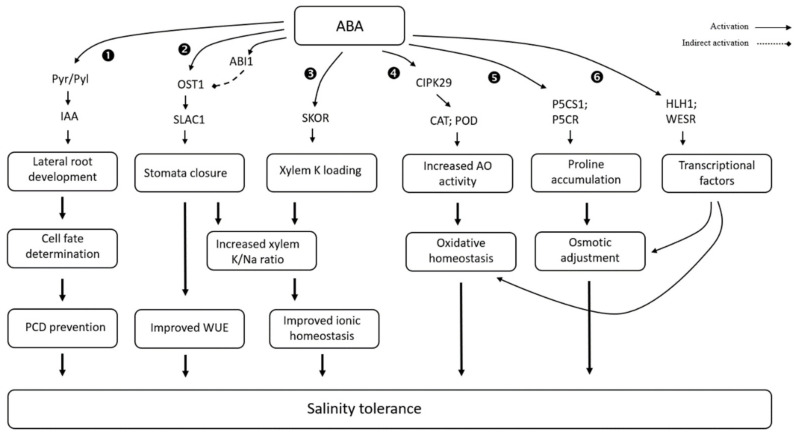
Effect of abscisic acid on salt stress tolerance in plants (based on ❶ [240], ❷ [223], ❸ [241], ❹ [236], ❺ [231], and ❻ [239]).

## Data Availability

Not applicable.

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
