# Peer review of "Improving Performance of Salt-Grown Crops by Exogenous Application of Plant Growth Regulators"

_biomolecules, 2021, doi:10.3390/biom11060788_

Round 1

Reviewer 1 Report

This review summarizes the effectiveness of various PGRs in ameliorating detrimental effects of salinity on plant growth and development and provides some insights into physiological and genetic mechanisms underlying this process.

  • Abstract is not clear. Please improve it.
  • Background of the study should be made to very clear. Provide more details of introduction and review of the work.
  • Please provide mechanism drawing figures. Please provide 2 figures and explain the mechanisms.
  • In Conclusion, the authors should add the significance of this research, perspectives and potential practical application.

Author Response

Comment: Abstract is not clear. Please improve it.

Reply: accepted. The Abstract was rewritten to include more specific details.

Comment: Background of the study should be made to very clear. Provide more details of introduction and review of the work.

Reply: accepted. The Introduction section was extended to provide more arguments of why PGR may be more attractive to plant growers, and what are the possible limitations (line 77-101).     

Comment: Please provide mechanism drawing figures. Please provide 2 figures and explain the mechanisms.

Reply: accepted. Following this request, three figures were added to the MS depicting effects of some PGR on downstream targets (genes/transporters).

Comment: In Conclusion, the authors should add the significance of this research, perspectives and potential practical application.

Reply: accepted; the requested information have been added to the conclusion section (line 556-585).

Reviewer 2 Report

The review work did not organaized well and it needs lot important information regarding methods for exogenous application of plant growth regulators. Authors just focussed on summary of others work. They did not discussed any of their own concerns and future implications which are important for review.

Author Response

Comment: The review work did not organaized well and it needs lot important information regarding methods for exogenous application of plant growth regulators. Authors just focussed on summary of others work. They did not discussed any of their own concerns and future implications which are important for review.

Reply: we partially agree with the Reviewer that our review is largely descriptive. However, we believe that the level of specific details (e.g. downstream molecular targets for PGR and possible signalling pathways) will be of a certain interest to journal’s readers. In the light of this criticism, we have also undertaken more critical assessment of the existing data (line 141-143; 168-170; 247-249; 273-280; 349-350 and 454-457). We have also significantly improved the last (conclusions/prospects) section by providing a comprehensive assessment of technological limitations and the cost-benefit rationale of PGR application (line 556-585). We hope that such modification will eliminate the Reviewer’s concerns.  

Reviewer 3 Report

1. This review is comprehensive and critical to the related field, some minor grammar errors should be corrected before publication. A paragraph describing the selection procedure of references used in this review is needed.

2. line 21: effectiveness of various PGRs - the effectiveness of various PGRs

3. line 23: while each of these PGRs possess - while each of these PGRs possesses

4. line 28: timing of application - the timing of application

5. line 32: A further progress - Further progress

6. line 46: reducing plant’s ability - reducing the plant’s ability

7. line 58: signalling - signaling (check it throughout the manuscript)

8. line 61: by human population - by the human population

9. line 66:  Australian wheatbelt -  Australian Wheatbelt

10. line 71: and include - and includes

11. line 92:  thephysiological -  the physiological

12. line 95: at nontoxic level - at a nontoxic level

13. line 330: to mitigate of salt stress damage - to mitigate salt stress damage

14. line 332: when plants expose - when plants exposed

15. line 346: In Arabidopsis - In Arabidopsis,

16. line 499: cost-efefctive - cost-effective

17. line 509: allow to quantify - allow quantifying

18. Graphic presentation or comprehensive tables could enhance the attractiveness to the readers.

Author Response

Comment: 1. This review is comprehensive and critical to the related field, some minor grammar errors should be corrected before publication.

  1. line 21: effectiveness of various PGRs - the effectiveness of various PGRs
  2. line 23: while each of these PGRs possess - while each of these PGRs possesses
  3. line 28: timing of application - the timing of application
  4. line 32: A further progress - Further progress
  5. line 46: reducing plant’s ability - reducing the plant’s ability
  6. line 58: signalling - signaling (check it throughout the manuscript)
  7. line 61: by human population - by the human population
  8. line 66: Australian wheatbelt - Australian Wheatbelt
  9. line 71: and include - and includes
  10. line 92: thephysiological - the physiological
  11. line 95: at nontoxic level - at a nontoxic level
  12. line 330: to mitigate of salt stress damage - to mitigate salt stress damage
  13. line 332: when plants expose - when plants exposed
  14. line 346: In Arabidopsis - In Arabidopsis,
  15. line 499: cost-efefctive - cost-effective
  16. line 509: allow to quantify - allow quantifying
  17. Graphic presentation or comprehensive tables could enhance the attractiveness to the readers.

Reply: accepted and fixed.

Comment: A paragraph describing the selection procedure of references used in this review is needed.

Reply: we do not believe this information should be included in the MS per se, as it was done using the standard search tools, For the reviewer’s benefits, some details are given below:

  • A standard literature search was performed using the world largest academic citation databases i.e., Scopus, PubMed, Web of Science and Google scholar.
  • The search was set from the beginning of the relevant article until 2021.

The following keywords were used as a search query: 1) plant growth regulator, 2) phytohormone, 3) PGRs, 4) plant hormones 5), salt stress tolerance, and 6) Auxin, cytokinins, nitric oxide, brassinosteroids, gibberellins, salicylic acid, abscisic acid, jasmonates and ethylene.  

Round 2

Reviewer 1 Report

Requested corrections were competed. 

I found some botanical names that were not italicized in the text and references. Botanical name should be italics. 

Reviewer 2 Report

Authors improved the MS as suggested. Now, i am accepting the MS for publication in Biomolecules Journal.